# Paralyzed by Fear?—A Case Report in the Context of Narrative Review on Catatonia

**DOI:** 10.3390/ijerph191610161

**Published:** 2022-08-16

**Authors:** Karina Badura Brzoza, Patryk Główczyński, Michał Błachut

**Affiliations:** Department of Psychiatry, Faculty of Medical Sciences in Zabrze, Medical University of Silesia, 40-055 Katowice, Poland

**Keywords:** catatonia, catatonia treatment, case study, psychiatry

## Abstract

In ICD-11, catatonia is a complex syndrome that includes psychomotor disorders (negativity, catalepsy, wax flexibility, mutism, automatism, mannerisms, or echolalia) and volitional processes affect modulation and action planning, which leads to hypofunctional, hyperfunctional, or parafunctional motor action. This is a very important clue that this state can be associated with both mental and somatic diseases. In order to create a narrative review, authors analyzed the diagnostic criteria of ICD-10 and ICD-11 and searched the PubMed medical base for articles on the diagnosis and different approaches to the treatment of catatonia. The treatment of catatonia is not standardized. It is based on the use of benzodiazepines, GABAa receptor antagonists, NMDA receptor antagonists, D2 receptor antagonist, and electroconvulsive therapy (ECT). The authors also would like to present the case of a patient in whom the diagnosis of catatonia was not so clear according to the diagnostic criteria, emphasizing the importance of the key diagnosis for the patient’s recovery. The authors would also like to point out that the topic of catatonia should be of interest not only to psychiatrists, but also to doctors of other specialties, who may encounter cases of catatonia complicating somatic states in hospital wards.

## 1. Introduction

Catatonia (Cata-completely; tonikos-tense) is a rare but spectacular psychomotor disorder that occurs in the course of both mental and somatic diseases. Catatonia (following ICD-11) is defined as a syndrome of primarily psychomotor disturbances, characterized by the occurrence of several symptoms of decreased, increased, or abnormal psychomotor activity. The complexity of this medical issue is evidenced by the fact that in the ICD-11 classification it has the rank of a subchapter, while in the ICD-10 it is presented in two ways—as organic catatonic disorder, F06.1, or catatonic schizophrenia, F20.2. The ICD-11 classification emphasizes the interdisciplinary nature of catatonia and even identified a new subtype—secondary catatonic syndrome—which can be observed in many different somatic diseases [1]. In the American DSM-III classification, cases of catatonia are classified as “catatonic-type schizophrenia”. The awareness that catatonia may be a syndrome independent of schizophrenia and pharmacotherapy is sometimes conducted with drugs that are ineffective in alleviating the symptoms of schizophrenia led to the development of diagnostic scales or provocation tests in order to correctly verify the diagnosis and develop effective treatment algorithms. In the DSM-IV classification, catatonia was identified as a syndrome secondary to the disease state, while in DSM-V, the catatonic schizophrenia class was removed, and catatonia was recognized as a distinct, verifiable, and treatable disease syndrome. Of course, catatonia can accompany many mental illnesses—the vast majority (43%) is bipolar disorder (BD), 30% in schizophrenia, there were also individual cases of PTSD syndrome, during withdrawal of benzodiazepines or alcohol, and in obsessive-compulsive disorders [2,3,4]. Catatonia was also observed in autism spectrum patients [5]. Moreover, according to scientific studies, more than a quarter of catatonia cases may be caused by a somatic condition, especially neurological disorders such as epilepsy, neuroinfections, autoimmune diseases [6], or encephalitis with anti-NMDA-r antibodies (NMDA receptors) [7]. In adolescent patients, catatonia may also coexist with another qualitative movement disorder, tics, but also with Wilson’s disease, porphyria, or, for example, MELAS syndrome, ketoacidosis, homocystinuria, hypercalcemia, neoplasms, hepatic encephalopathy, or encephalitis [8,9]. In order to create a narrative review, the authors analyzed the diagnostic criteria of ICD-10 and ICD-11 and a review of PubMed databases. The criteria for the selection of literature were to take into account the publications from 2013–2022 presenting methods of treating catatonia, but also including historical works from the last century in order to emphasize the differences and dynamically change the approach. Moreover, a clinical case of a patient was described in order to present the therapeutic difficulties that arise in the course of medical care for patients diagnosed with catatonia.

## 2. Historical View

Catatonia, or rather its components, have been known to mankind since its inception. Perhaps catatonic symptoms indirectly inspired the creators of the myth about Medusa, one of the Gorgons, who through her eyes immobilized the mortals who went to her place [10]. The disease itself was initially described with various terms—catalepsy, hysteria, nostalgia syndrome (in people far away from home, usually soldiers), and could be an element of medieval asceticism (which also included immobility or refusal to eat) or even read as a possession. The components of catatonia—immobility, stupor, waxy flexibility, mutism, negativity, omission of food and fluids—undeniably belong to quite spectacular symptoms, which gave a paranormal, magical meaning, generated anxiety, and sometimes led to the recognition of the person as dead. It is worth emphasizing, however, that what people regarded as “demon possession” has almost always been perceived in medicine as a group of various diseases with a different—not fully understood—etiology. The term catatonia was first used by Kahlbaum in 1874, distinguishing it as a separate disease entity associated with organic brain damage [11,12]. A quarter of a century later, Emil Kraepelin classified catatonia as a subtype of this disease (dementia praecox) on the basis of his conception of schizophrenia. This fact had two consequences—the concept of catatonia has not been forgotten over the years, and at the same time, for a long period of time, it has become mistakenly associated with the diagnose of schizophrenia. The enormous authority of Kraepelin meant that the dogmas introduced by him survived for the next hundred years, cultivated by his successors [13]. At the beginning of the 20th century, Kraepelin’s approach was supplemented and modified by Bleuler—he departed from the diagnosis of dementia praecox in the course of catatonia, pointing out that a person can develop it at any age, but the main assumption of a permanent link between catatonia and schizophrenia remained unchanged. There have been many difficulties, and this still raises the question of what is going on in the minds of patients who develop catatonia. Two beliefs dominated in this aspect. Some researchers claimed that these minds were then empty, almost excluded from thought processes. The other (also called the School of Anxiety and Fear and associated with it—Karl Leonhard), which over time became the dominant trend, believed that the patient was overwhelmed with fear and anxiety, pointing out that the presence of major stressors could lead to catatonia. Following Karl Leonhard, it could be assumed that severe anxiety can cause catatonia and is the dominant feeling in the patient’s mind. Along with the thesis of anxiety, there was a theory of stupor, where a patient in catatonia would be in a specific “analgesia” [14]. The effectiveness of benzodiazepines (understood as anti-anxiety drugs) may confirm the views of Leonhard’s school. Confirmation of the hypothesis related to the presence of paralyzing anxiety in catatonia can also be found in the observation of the behavior of animals and their freezing (tonic reaction) in motionlessness in response to a threatening danger. It is impossible not to mention the fact that the symptoms of catatonia appear in the course of diseases associated with severe anxiety, such as schizophrenia, affective diseases, or PTSD, but also individuals that could be catatonic, such as possession, homesickness, or even asceticism [15]. Cuevas-Esteban came to other conclusions, in a study in which he showed a correlation between the inhibition domain and the occurrence of catatonia, which prompted him to propose that not every appearance of catatonia symptoms is associated with high anxiety [16].

## 3. Etiology, Components, and Scales of Catatonia

Catatonia is a complex syndrome that includes psychomotor disorders (negativity, catalepsy, wax flexibility, mutism, automatism, mannerisms, or echolalia) and volitional processes to affect modulation and even action planning, which leads to reduced (hypofunctional), excessive (hyperfunctional), or abnormal (parafunctional, parakinesis) motor action. It is worth emphasizing that some of the symptoms may increase significantly in the presence of doctor examining the patient. To diagnose catatonia (DSM-V), at least three of the listed dysfunctions must be present: stupor, catalepsy, waxy flexibility, negativism, posturalism, mutism, agitation, stereotypes, mannerisms, echolalia, echopraxia, and grimacing [17]. Searching for the etiology of catatonia, the vast majority of researchers focused mainly on motor symptoms as the dominant aspect of the disorder [18]. They have even developed models of neural circuits for motor abnormalities in various psychiatric conditions. Research on schizophrenia has shown some analogies between catatonia and parkinsonism or other dyskinesias [19]. On the other hand, referring to the history of psychiatry, Kraepelin already drew attention to volitional disorders as a starting point for catatonia [20], while Bleuler considered the presence of negativity and ambivalence as fundamental as the basis for the disappearance of conscious behavior [21]. Although the pathophysiology of catatonia (as emphasized by the completely heterogeneous syndrome) remains unknown, it is suggested to adopt the thesis about the dysregulation of glutamatergic, dopaminergic, and GABAergic transmission [22]. Apart from the activity of neurotransmitters itself, attention is drawn to the disturbed number of GABA receptors (GABAr) in the sensorimotor cortex of people showing catatonic symptoms [23]. Less popular are the hypotheses about the dysregulation of the autonomic system. [24]. The course of acute catatonia mimicking, in some cases, meningitis and encephalitis, gave rise to a hypothesis of disorders in the immune system [25,26]. There are few scales to help to diagnose catatonia, the Bush and Peralta being the most used. The oldest one is BFCRS—The Bush–Francis Catatonia Rating Scale. That was an instrument constructed for the standardized examination of catatonia using defined symptoms. Unfortunately, the number of signs from BFCRS needed to diagnose catatonia remains unclear. The second one is Peralta–Cuesta scale PANSS, the Positive and Negative Syndrome Scale, which contains a positive scale, negative scale, and general psychopathology scale [27]. One of the newer, but also more accurate, scales is the CS (Catatonia Spectrum) scale. It is characterized by high internal consistency, positive convergent validity with alternative dimensional measures of catatonia and sound test-retest reliability. It consists of 74 items and is divided into 8 domains: Stupor; Mutism; Stereotypes; Mannerisms; Negativism; Automatic obedience; Automatisms and Impulsivity. For each item there is a dichotomous answer “Yes” and “No” [28].

## 4. Treatment of Catatonia

The breakthrough step for the understanding of catatonia was the introduction of strong antipsychotic drugs and the subsequent emergence of neuroleptic malignant syndrome (NMS), associated with mutism, stupor, fever, and tachycardia as a result of their use. It was then that the blockade of dopaminergic receptors was associated with the above symptoms, but the results were mediocre in the use of agonists of these receptors as a specific antidote. The association of NMS with the notion of catatonia led to the appearance of preliminary postulates to distinguish catatonia as a separate entity and initiated the resumption of the discussion on the reclassification of Kraepelin. Catatonia was observed more and more often in the course of depression than in schizophrenia. In the late 1980s, James Lohr and Alexander Wiśniewski pointed to the independence of the diagnosis of catatonic syndrome, citing evidence that it is a separate entity, but should be diagnosed solely on the basis of motor symptoms. The need to reclassify catatonia was provided by reports of deaths as a result of treatment with strong neuroleptics [29]. Initially, all therapy trials were used with little success, until in 1930, amobarbital proved to be effective in the treatment of catatonia. Induced seizures (used as a treatment for psychosis) have also been found to be effective in reducing catatonia. These methods turned out to be so effective that the incidence of catatonia has decreased to such an extent that it was wondered whether catatonia was no longer a medical challenge [30]. In the 1980s, benzodiazepines, initially used as barbiturate substitutes, slowly displaced them from treatment. The effectiveness of benzodiazepines in alleviating almost 80% of catatonia cases, with the success of electroconvulsive treatment (ECT) in the remaining cases, favored the belief that catatonia should be understood as a separate and diverse disease.

### 4.1. Benzodiazepines (BDZ)

Undeniably, benzodiazepines are the first line of drugs used in catatonia, regardless of its etiology. Their mechanism of action is based on the allosteric regulation of GABA receptors in the frontal cortex, and thus GABAergic correction in the patient’s neural networks. In trials of catatonia therapy, a remission rate of up to 80% was recorded, without showing a direct relationship between increasing the dose and the speed of the effect—although the use of higher than recommended doses did not cause more frequent adverse effects [31]. Although there are no specific recommendations for a specific benzodiazepine substance commonly used in the treatment of catatonia, lorazepam is the most popular (79% remission), and the available literature also proves the effectiveness of oxazepam, clonazepam, and diazepam [27]. Unfortunately, there are no clear guidelines as to the period of use of BDZ after the symptoms of catatonia have subsided—clinical experience, however, suggests the need to continue treatment for a longer time after the episode has subsided [32]. Many researchers indicate the need to dose lorazepam initially 1–2 mg every 4–12 h, with a subsequent increase in the dose to 8–24 mg daily. With this procedure, you can count on clinical improvement within a week, but this time may be extended individually [33]. Benzodiazepines are also used in the catatonia diagnostic test [34].

Certain factors may influence the efficacy of benzodiazepines in the pharmacotherapy of catatonia. The longer the time between the onset of symptoms and starting treatment, the worse the response to benzodiazepines [35]. The appropriate dosage is also important, because on this basis the effectiveness is also determined (with particular emphasis on the doses not being too low). Chronic catatonia in the course of schizophrenia is, in many cases, resistant to pharmacotherapy [36].

### 4.2. GABA_A_ Receptor Alpha Agonists

Among the drugs with this mechanism of action, zolpidem has found widespread use, which was first investigated by Mastain et al. in 1995. However, its effectiveness is related to the need to monitor the therapeutic concentration in blood serum, ranging between 80 and 130 ng/mL [37]. The same authors described in the Lancet cases of recurrence of symptoms after the concentration dropped below 90 ng/mL [38]. Currently, it is considered in situations where catatonia turns out to be resistant to benzodiazepines and electroconvulsive therapy in both acute and long-term treatment [39]. Zolpidem may also be an alternative to benzodiazepines in the catatonia diagnostic test [40].

### 4.3. Electroconvulsive Therapy (ECT)

The first ECT treatments for the therapeutic purposes of catatonia started to be used in the 1940s, after previous attempts at treatment with amobarbital. Even then, the effectiveness of ECT therapy was noticed, however, treating it as an “alternative” method of treatment putting pharmacotherapy in the first place. Perhaps it was related to the very availability of benzodiazepines/barbiturates or the unavailability of electroconvulsive therapy centers. The current knowledge, however, clearly indicates the need to consider ECT as a priority, especially in situations where catatonia is severe (“malignant catatonia”—co-occurring with tachycardia, blood pressure changes, and fever), when it may be associated with an affective disorder, or when it is already characterized by initial lorazepam resistance. In these cases, the effectiveness of electrotherapy is 80–85% [41], although there are no randomized studies on this subject. Suzuki et al. achieved an astonishing 100% improvement in the group of patients diagnosed with catatonic schizophrenia. Unfortunately, in 60% of patients symptoms recurred despite continuing pharmacotherapy, which may indicate the need for repeat ECT [42]. Many researchers also demonstrated high effectiveness of ECT after previous ineffective pharmacotherapy with benzodiazepines [43]. In Payee’s studies, almost 90% of patients responded positively to ECT procedures performed after previous ineffective lorazepam therapy [44,45]. Unfortunately, most studies are based on a small number of patients, which may be associated with a low incidence of catatonia. There are also no specific guidelines as to the number of ECT sessions needed to achieve clinical improvement, hence the need to individualize the approach tailored to the condition of a particular patient [46]. Combining benzodiazepines and ECT is also a significant challenge. It is permissible to use lorazepam together with electroconvulsive sessions when discontinuation of the previously used benzodiazepine could worsen the patient’s condition. However, when lorazepam hinders the induction of an effective epileptic seizure, it is recommended to use flumazenil before anesthesia for ECT [35].

### 4.4. NMDA Receptor Antagonists

Amantadine (and memantine) have been considered by few researchers as potentiators of the effects of ECT [47], as well as of benzodiazepine pharmacotherapy [48], and also considered for use as monotherapy [49]. Unfortunately, the amount of evidence and its quality do not support their routine use, but it does offer some prospects for the continuation of research on this group of drugs.

### 4.5. D2 Receptor Antagonists

Despite the quite unambiguous position regarding the use of antipsychotic drugs blocking the D2 receptor, which says that they should be discontinued as early as possible during the onset of catatonia symptoms due to the risk of exacerbation of symptoms, there are publications indicating their beneficial effect in specific situations. Pharmacotherapy with second-generation drugs (weak antagonists such as quetiapine or partial D2 agonists such as aripiprazole) applied after relieving the symptoms of catatonia may be helpful in combating psychotic symptoms, or as their prophylaxis [50].

### 4.6. Repetitive Transcranial Magnetic Stimulation (rTMS)

There is much hope for the use of rTMS in cases of catatonia, which pharmacotherapy turns out to be ineffective and safety considerations do not allow the use/continuation of ECT [51]. We purposely use the term “hope” here, because despite the documented effectiveness of the method, the prevalence of centers using rTMS is very low [52].

In the context of these considerations, a clinical case of a patient with catatonic manifestations was presented along with the therapeutic process.

## 5. Case Report

An 18-year-old patient was treated from the age of 15 with a diagnosis of bipolar disorder and Asperger’s syndrome. The first symptoms of the disease appeared during a trip to Egypt with his parents in 2016. The patient had trouble sleeping, and became restless, irritable, and illogical in his statements. He remained in symptomatic remission for a long time after systematically treating at the Children’s Mental Health Clinic, taking aripiprazole at a dose of 20 mg/day. In April 2019, the attending psychiatrist, taking into account the absence of disease symptoms and the patient’s good functioning, together with the patient and his parents, decided to end pharmacotherapy. In July 2019, the patient developed sleep problems, became restless, and began to withdraw from social contacts. Outpatient treatment with the same drug was reintroduced, but it did not bring improvement. The patient was consulted psychiatrically at the admission room, where acute psychotic disorders were diagnosed, but admission to the ward was not decided. A few weeks later, due to his constantly deteriorating mental state, the patient was sent to a psychiatric ward. During the admission, the patient was in poor verbal contact, with clear awareness, autopsychically oriented, poorly allopsychically oriented, in limited emotional and intellectual contact, he gave a psychotic impression, however, full assessment of psychotic experiences was not possible due to significantly limited verbal contact with the patient. From the beginning of hospitalization, he refused food and fluids, and was extremely tense, autistic, and scared. During the attempts to perform nursing activities, he resisted and reacted aggressively. On the second day of hospitalization, he needed to be secured with magnetic belts due to increasing psychomotor agitation and aggressive gestures. The symptoms presented by the patient (stupor, periodic sudden agitation, negativity, mutism) became the basis for the initial diagnosis of catatonic schizophrenia and the initiation of treatment. Diazepam and levopromazine injections and parenteral hydration were included in the treatment. The patient’s condition did not improve. He remained motionless most of the time, and showed negativity when administered drugs and care. Most time he was mute, and made only rudimentary verbal contact, and only with his parents. Moreover, the patient appeared to be in a strong anxiety, which increased in the presence of strangers and presented with intense negativity and a terrified expression on his face, and slightly decreased symptoms in the presence of his parents. On the fourth day of hospitalization, anticoagulation was started, the patient was catheterized, and the basic treatment was changed to aripiprazole injections. On the fifth day of hospitalization in the afternoon, the patient’s somatic condition suddenly deteriorated, symptoms of cyanosis, dyspnea with a decrease in saturation, tachycardia, lower blood pressure, and quantitative disturbances of consciousness appeared. The patient was transferred to the Intensive Care Unit with the diagnosis of acute respiratory failure in the course of massive pulmonary embolism. In the ward, the patient was intubated and treated with thrombolysis. While still in the Intensive Care Unit, after disconnecting from the respirator, antipsychotic treatment was started (aripiprazole at a dose of 7.5 mg), and a CT scan of the head performed excluded signs of CNS bleeding. After stabilization of the somatic state on the seventh day of hospitalization, the patient was again transferred to the psychiatric ward. On admission, the patient was without verbal contact, made eye contact, reacted to voice and tactile stimuli, did not make deliberate movements, showed passive negativity when trying to change position, an accurate assessment of psychotic symptoms was not possible, the patient reacted with a strong fear to the sight of all persons except parents. From the beginning of hospitalization, the patient was unable to consume food, fluids, and medications orally. Fed on a specialized industrial diet through an enteral tube, he was irrigated parenterally. Antipsychotics—aripiprazole, amisulpride, and lorazepam—were administered via a gastric tube. Antibiotic therapy was continued, and a catheter was kept in the bladder. Mental state was slowly improving, and the patient began to establish simple verbal contact with his parents and remembered the events of the last days. He showed enormous fear and distrust towards the medical staff. Physical rehabilitation began only in company of parents, and he administered it passively. After several days, the patient’s somatic condition began to deteriorate again, a high fever appeared. The patient was referred to the Department of Internal Medicine with suspicion of urosepsis. After stabilization of his somatic condition, he was again transferred to a psychiatric ward, this time to another psychiatric hospital, for electroconvulsive therapy. On admission, the patient was without verbal contact, occasionally made eye contact, negativistic, put up active resistance during the physical examination, and did not appear to be hallucinating. Initially, he refused to take food and fluids, and was fed through a gastric tube. Pharmacotherapy with aripiprazole 30 mg daily was slowly discontinued during one week after admission and lorazepam was increased from 2 mg daily to 7.5 mg. Eventually, the dose was set to 10 mg daily. In the ward, the urinary tract was re-infected (Pseudomonas Aeruginosa) and antibiotic therapy was successfully applied. Mental condition began to slowly improve. Despite the original assumption, no decision was ultimately made to start ECT treatments. Taking into account the previous massive pulmonary embolism, sharp parental opposition, and the slow but clearly noticeable improvement in mental state, the decision to start ECT was postponed, and the pharmacotherapy was changed. On the 18th day of hospitalization, taking into account the patient’s improving state and earlier medical history suggesting a diagnosis of catatonic schizophrenia, it was decided to start the atypical neuroleptic—olanzapine. The dose was gradually increased from 5 mg to the final dose of 15 mg per day, constantly maintaining the lorazepam treatment and then gradually reducing it to 7.5 mg per day on the day of discharge. The patient was discharged on the 53rd day of hospitalization, calm, adjusted in behavior, in a balanced drive and mood, did not express delusions or hallucinations and denied the occurrence of suicidal thoughts. Formal thinking disorders, such as verbosity, periodic paralogy, and cognitive impairment, gradually subsided. However, blunted affect and impoverishment of thinking was still observed. It was recommended to maintain lorazepam at a dose of 7.5 mg orally in a divided daily dose (for subsequent reduction at the Mental Health Clinic) and olanzapine in a 15 mg donor. At the moment, the patient is functioning properly, he has graduated his high school and is constantly treated on an outpatient basis. He is in good verbal contact, with no formal disturbance of thought and psychotic symptoms. Pharmacological treatment, due to side effects occurring after olanzapine (cognitive impairment, weight gain), was changed back to aripiprazole at a dose of 15 mg per day.

## 6. Discussion

Catatonia is a syndrome that is still life-threatening and requires intensive treatment [53]. The available data show that catatonia is more common among adolescent schizophrenic patients compared to the group of adults suffering from schizophrenia [54].

The aim of our study was to draw attention to the possible occurrence of catatonic syndrome symptoms in a patient diagnosed with autism spectrum disorders and bipolar disorder (primary diagnosis), as well as to present a severe course of a catatonic episode, complicated by somatic disorders, in the described case. The deterioration of the somatic state associated with pulmonary embolism and then urosepsis could be a factor that aggravated catatonic symptoms and hindered effective therapy. The somatic condition was also a factor that caused that, despite the proven high effectiveness of ECT treatments in the treatment of catatonia. In the case of the above-mentioned patient, a decision was finally made to conduct pharmacological treatment with a positive effect. The authors also would like to emphasize, in narrative review, the importance of the fact that catatonia may or may not be caused by a mental disorder. Therefore, the knowledge about catatonia is important for doctors of every specialty. Even though it is considered a rare disease, it can happen, and early diagnosis of catatonia can be crucial for a patient.

## 7. Conclusions

Catatonia is a systemic disease, not just a mental disorder in which progressive, increasing anxiety may be a contributing factor.The pathomechanism of catatonia, though undoubtedly complex, is still unknown.In the treatment of catatonia, drugs from the benzodiazepine group and electroconvulsive treatments are mainly used. Moreover, attempts are made to use agonists of the GABA_A_ subunit and, less frequently, NMDA antagonists.Treatment of catatonia should be as individual as possible and tailored to the patient’s needs.

## Data Availability

Not applicable.

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
