# Peer review of "Paralyzed by Fear?—A Case Report in the Context of Narrative Review on Catatonia"

_ijerph, 2022, doi:10.3390/ijerph191610161_

Round 1
Reviewer 1 Report
The paper is an interesting case study enriched by introduction and discussion. Topic is important, nevertheless some issues need to be adressed
1. Title: "Paralyzed by fear (...)" refers to one of the catatonia conceptualizations. The topis is mentioned in the manuscript, but not explored enought to justify such a title
2. Abstract. The Authors inform that they "analyzed the diagnostic criteria of ICD-10 10 and ICD-11". The ICD-10 and 11 criteria are mentioned in the text along with DSM conceptualization. however no conclusion is drawn regarding to the described patient
3. Abstract. The Authors inform that they "reviewed the PubMed medical base". What was the literature search protocol?
4. Introduction. Although the Authors mention numerous catatonia conceptualizations, actually the disorder definition is lacking. The Authors just briefly mention possibility somatic complications, which are the important part of the case study description
5. Case study. The primary and final diagnoses of the patient are not provided. I believe that the patient was catatonic. However, the described symptoms does not fit to the DSM or ICD criteria - for example food or drug refusal are not catatonia symptoms. The patient's state description should be revised and observed symptoms enlisted leading to some diagnostic conclusion.
6. If the patient was catatonic, it is unclear why he was all the time administered antipsychotics. What symptoms those drugs were expected to target?
7. Why antipsychotics were started just after the severe pulmonary embolism while it is known that this group of drug is associated with increased embolism risk and no acute psychotic symptoms were present?
8. So severe somatic complications, although possible, are not common. Actually do Authors have some hypotheses why this happened?
Author Response
Dears Sirs
Thank you for revising article entitled “Paralyzed by fear? - a case report in the context of narrative review on catatonia.” We have reorganized the paper following Reviewers’ comments. All changes in the text of article have been written in red pencil. There is our respond.
The paper is an interesting case study enriched by introduction and discussion. Topic is important, nevertheless some issues need to be adressed
- Title: "Paralyzed by fear (...)" refers to one of the catatonia conceptualizations. The topic is mentioned in the manuscript, but not explored enough to justify such a title
We changed the title of article by adding the question mark after the fear - „Paralyzed by fear? - a case report in the context of narrative review on catatonia”.
2. Abstract. The Authors inform that they "analyzed the diagnostic criteria of ICD-10 10 and ICD-11". The ICD-10 and 11 criteria are mentioned in the text along with DSM conceptualization. however no conclusion is drawn regarding to the described patient.
Authors changed the abstract. We added ICD-11 definition, mentioned why we searched for articles in PubMed. Also we added why we did the revision in the case of our patients - to present person with not so clear diagnostic criteria, emphasizing the importance of the key diagnosis for the patient’s recovery. We also add note about importance knowledge about catatonia for doctors of every specialty, because even it is consider as a rare disease, it can happen, and early diagnosis of catatonia can be crucial for a patient.
3. Abstract. The Authors inform that they "reviewed the PubMed medical base". What was the literature search protocol?
We added literature search protocol into Introduction.
4. Introduction. Although the Authors mention numerous catatonia conceptualizations, actually the disorder definition is lacking. The Authors just briefly mention possibility somatic complications, which are the important part of the case study description
The authors added the definitions of catatonia based on the ICD-11, and additionally extended the somatic causes (more popular ones) of catatonia.
5. Case study. The primary and final diagnoses of the patient are not provided. I believe that the patient was catatonic. However, the described symptoms does not fit to the DSM or ICD criteria - for example food or drug refusal are not catatonia symptoms. The patient's state description should be revised and observed symptoms enlisted leading to some diagnostic conclusion.
The primary and final diagnosis was provided. The patient state description was modified.
6. If the patient was catatonic, it is unclear why he was all the time administered antipsychotics. What symptoms those drugs were expected to target?
Treatment with antipsychotic drugs was started in the previous psychiatric ward. After the patient was admitted to our hospital, aripiprazole was gradually discontinued, keeping the patient treated only with lorazepam. During hospitalization, his mental state improved, the patient began to establish verbal contact and move independently. Nerveless some paralogical thinking, blunted affect, and weirdness were present in the psychiatric examination. Taking into account the previous history of the disease and the symptoms present during the current hospitalization, the diagnosis of catatonic schizophrenia was finally set and treatment with olanzapine was introduced, which is continued until now.
7. Why antipsychotics were started just after the severe pulmonary embolism while it is known that this group of drug is associated with increased embolism risk and no acute psychotic symptoms were present?
We fully agree with the fact that antipsychotic drugs increase the risk of embolism, unfortunately we are not able to answer this question because this treatment was started in the previous psychiatric ward before the patient was admitted to our hospital.
8. So severe somatic complications, although possible, are not common. Actually do Authors have some hypotheses why this happened?
There are actually a few hypothesis for the state of the patient. Pulmonary embolism could be caused by insufficient anticoagulation (not adjusted to body weight) or introduced too late, he was lying in his bed for some days. Maybe he has some thrombotic disease (for example Antiphospholipid syndrome?) Urosepsis could be caused by a catheter which was kept in the bladder (maybe too long, or maybe - the procedure of placing the catheter was not sterile enough?). Very important information is connecting with Intensive Therapy Unit, where many patients can get some bacterial infection.

Reviewer 2 Report
The case report is very interesting and the review of the literature is precise and concise. We may regret the choice not to cite diagnostic evaluation tools (Bush and Peralta scales for example) which is not an obstacle to publication as such.
Author Response
Thank you for revising article entitled “Paralyzed by fear? - a case report in the context of narrative review on catatonia.” We have reorganized the paper following Reviewers’ comments. All changes in the text of article have been written in red pencil. There are our responds to the Reviewer 2.
Dears Sirs
Thank you for revising article entitled “Paralyzed by fear? - a case report in the context of narrative review on catatonia.” We have reorganized the paper following Reviewers’ comments. All changes in the text of article have been written in red pencil.
The case report is very interesting and the review of the literature is precise and concise. We may regret the choice not to cite diagnostic evaluation tools (Bush and Peralta scales for example) which is not an obstacle to publication as such.
Authors added these information to etiology, components and scales of catatonia

Reviewer 3 Report
Dear Authors
Where there are many reviews on Catatonia, would you please mention why does one need to read your paper? What's unique?
What new we can learn from this ?
Paralysed by fear - good to mention the patient's experience
is it a review or case report?
If review what your criteria for inclusion or exclusion?
Author Response
Thank you for revising article entitled “Paralyzed by fear? - a case report in the context of narrative review on catatonia.” We have reorganized the paper following Reviewers’ comments. All changes in the text of article have been written in red pencil. There is our respond.
Where there are many reviews on Catatonia, would you please mention why does one need to read your paper? What's unique?
With this review, the authors wanted to present the current approach to catatonia, presenting both the historical outline, evolution and the definitions from the ICD-11. The authors also described a case report that was somatically complicated and may be one of the clues for others - not only psychiatrists, if they encounter a similar case. Moreover, the authors emphasize that catatonia can be caused by the patient's somatic condition and that knowledge about diagnosing catatonia should be popularized among physicians of all specialties.
What new we can learn from this ?
Unfortunately, so far there are no standards for the treatment of catatonia, but only guidelines. In the authors' opinion, each case described in the literature may be helpful for other psychiatrists when they encounter this disease entity. The described patient was complicated by somatic diseases, which is also emphasized by the somatic causative agent of catatonia, which is being discussed far too little.
Paralyzed by fear - good to mention the patient's experience
We changed the title of article by adding the question mark after the fear - „Paralyzed by fear? - a case report in the context of narrative review on catatonia”. We would like to stay with some conceptualization about catatonia.
is it a review or case report?
That is a case report in the context of review.
If review what your criteria for inclusion or exclusion?
We added these information to introduction - The criteria for the selection of literature was to take into account the publications from 2013-2022 presenting methods of treating catatonia, but also including historical works from the last century, in order to emphasize the differences and dynamically change the approach.

Round 2
Reviewer 3 Report
Good work